



# CSDMS: A community platform for numerical modeling of Earth-surface processes

Gregory E. Tucker[1,2], Eric W.H. Hutton[3], Mark D. Piper[3], Benjamin Campforts[3], Tian Gan[3], Katherine R. Barnhart[1,2,4], Albert Kettner[3], Irina Overeem[2,3], Scott D. Peckham[3], Lynn McCready[3], and Jaia Syvitski[3]

[1]Cooperative Institute for Research in Environmental Sciences (CIRES), University of Colorado Boulder, USA
[2]Department of Geological Sciences, University of Colorado Boulder, USA
[3]Institute for Arctic and Alpine Research (INSTAAR), University of Colorado Boulder, USA
[4]Current address: Landslide Hazards Group, US Geological Survey, Golden, Colorado, USA

**Correspondence:** G.E. Tucker (gtucker@colorado.edu)

**Abstract.** Computational modelling occupies a unique niche in Earth and environmental sciences. Models serve not just as scientific technology and infrastructure, but also as digital containers of the scientific community's understanding of the natural world. As this understanding improves, so too must the associated software. This dual nature—models as both infrastructure and hypotheses—means that modelling software must be designed to evolve continually as geoscientific knowledge itself
evolves. Here we describe design principles, protocols, and tools developed by the Community Surface Dynamics Modeling System (CSDMS) to promote a flexible, interoperable, and ever-improving research software ecosystem. These include a community repository for model sharing and metadata, interface and ontology standards for model interoperability, language bridging tools, a modular programming library for model construction, modular software components for data access, and a Python-based execution and model-coupling framework. Methods of community support and engagement that help create a
community-centered software ecosystem are also discussed.

## 1   Introduction

Our planet's surface is a dynamic place, changing on timescales from the momentary triggering of a landslide, to year-by-year resculpting of coastlines, to the formation of mountains and sedimentary basins over geologic time. The challenge of living sustainably on a dynamic, human-impacted planet is multi-faceted and multi-disciplinary, and requires a deeper understanding
of a diverse set of processes ranging from permafrost melting to wildfire impacts, and from river delta sinking to changes in flooding. These interwoven research challenges have two things in common: they cross traditional boundaries of research, and their solution requires computational models and model-data integration. Meeting these challenges efficiently requires an effective, integrated, and holistic software cyberinfrastructure to support computational modeling and analysis across the environmental sciences. Models embody theory in a quantitative and algorithmic form. By performing calculations at blinding
speed, numerical models extend our cognitive abilities, helping us explore and visualize the consequences of hypotheses.





They allow us to apply existing theory to new situations. Where the processes are sufficiently understood, models can forecast potential trajectories of natural and our anthropogenically perturbed Earth systems.

Creating, modifying, applying, and maintaining the software that implements numerical models requires time, money, and specialized skills. The software may be invisible, but its creation and maintenance constitute an infrastructure investment just as vital to science as the infrastructure supporting ship-based science or radio astronomy. More efficient infrastructure allows for more time devoted to other aspects of research and practice. And just as with laboratory infrastructure, scientific results that rely on software cyberinfrastructure are only as robust and reproducible as the software itself. Scientific software therefore needs quality control: errors in scientific software not only impede research, but also can produce misleading results that lead to more serious consequences. The fact that modeling is both useful and technically challenging can give rise to a pernicious temptation: to use an inadequate model for the job simply because the code that implements it is more easily available or more usable than better alternatives (Addor and Melsen, 2019).

A modular community software infrastructure must therefore maximize flexibility, creativity, and reliability while minimizing technical overhead. To use an artistic analogy: an ideal modeling infrastructure should provide the geo-artist with a wide palette of colors, while making it easy to mix new ones, so that more time can be devoted to creating, and less time to fussing with materials. Those materials must also be robust enough that the colors and textures will not degrade over time.

Here we describe software tools, standards, and practices that are designed to enhance research productivity by reducing the "time to science" in Earth modeling. Such tools and concepts form the key elements behind the Community Surface Dynamics Modeling System (CSDMS). Founded in 2007 with major support from the US National Science Foundation, CSDMS is a facility that supports and promotes computational modeling of diverse Earth-surface processes, in domains that span geomorphology, sedimentology, stratigraphy, marine geology, hydrology, and related aspects of geodynamics, geochemistry, soils, ecosystems, and human dimensions. CSDMS is currently organized into 12 community interest groups, representing about 2,000 members, and a small (about six full-time equivalent positions) Integration Facility that manages a web portal, develops middleware, and coordinates community events and resources. Here we present tools and standards developed by and for the CSDMS community. We describe a set of effective engineering practices that are well known among professional software developers but less known among geoscientists and environmental scientists. We highlight aspects of the human element: community engagement and education turn out to be key elements in forging a shared and ever-improving computational ecosystem.

We start with a background review of issues in scientific computing and research software across the sciences (Section 2), and a brief history of CSDMS (Section 3). Section 4 frames the operational tasks involved in numerical process modeling as a six-fold spectrum, ranging from simply executing a model program, to building a complete model from scratch. This sets the stage for a review of tools and practices designed to make these various tasks more efficient and their products more sustainable, through sharing, standardization, education, and a set of enabling tools (Sections 6–7). We conclude with a discussion of opportunities, needs, and challenges (Section 8).





## 2 Background

### 2.1 Scientific computing is here to stay

Computing has emerged as a pillar of scientific inquiry, alongside theory, experiment, and direct observation (PITAC, 2005). The ability to perform calculations at speeds that would have astonished researchers of our grandparents' generation continues to open up new territory across the sciences, and allows us to probe the limits to predictability in natural and engineered systems (Post and Votta, 2005; Post, 2013). Computing, and the software that supports it, underlies numerous recent success stories,

from improved hurricane forecasting to the imaging of black holes.

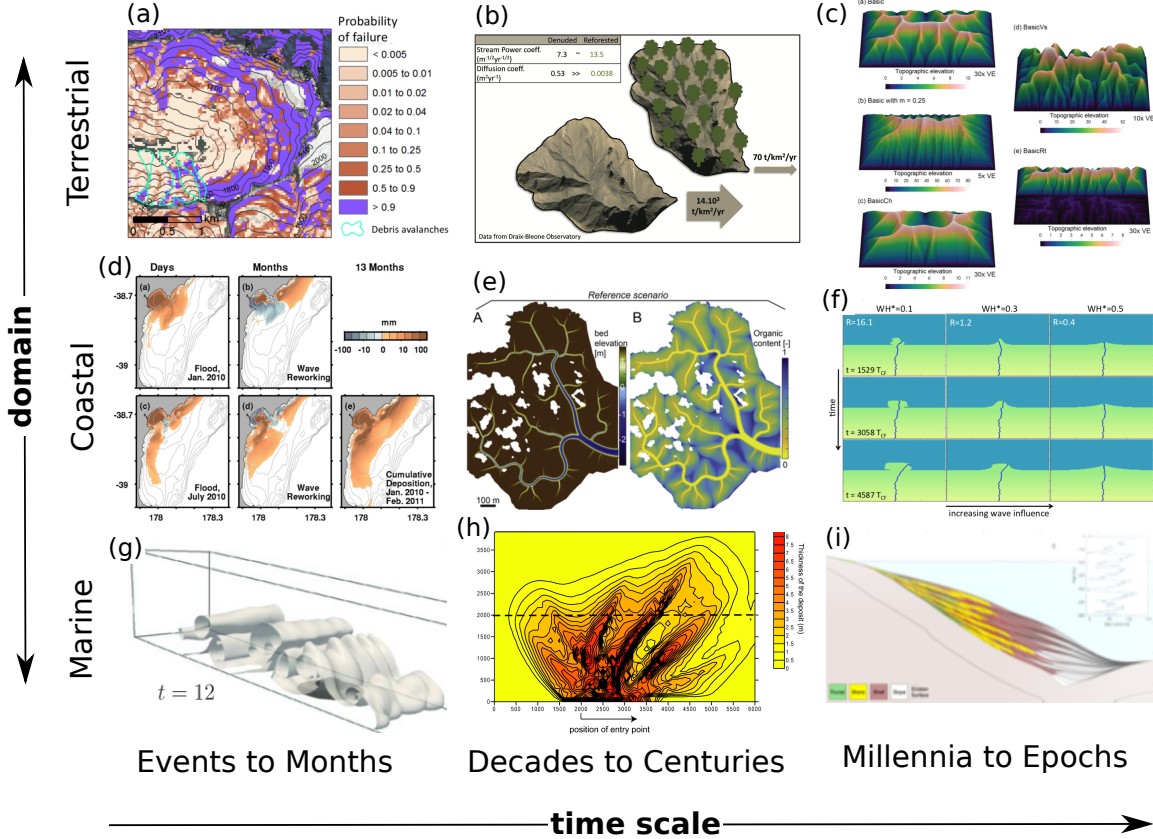

**Figure 1.** Examples of Earth surface process models across a variety of domains and time scales, here focusing on models of sedimentary processes. (a) Probabilistic landslide occurrence (Strauch et al., 2018). (b) Catchment sediment yield (Carriere et al., 2020). (c) Landform evolution (Barnhart et al., 2019). (d) Coastal and shelf dispersal of fluvially derived sediment (Kuehl et al., 2016). (e) Salt marsh evolution under tidal and sea-level forcing (Mariotti, 2018). (f) Delta evolution as a function of river and wave forcing (Ratliff et al., 2018). (g) Turbidity current dynamics (Nasr-Azadani et al., 2013). (h) Submarine turbidity fan stratigraphy (Groenenberg et al., 2010). (i) Sequence stratigraphy (Steckler et al., 2019).





Within the sphere of computing, numerical modeling—defined here as the computing of solutions to a set of equations and algorithms that represent a system—plays a central role. The process of formulating a computational model and the theory behind it encourages deep and precise thinking (e.g., Guest and Martin, 2020). Computational models both encapsulate theory, and provide machinery with which to explore the consequences of that theory. Pipitone and Easterbrook (2012), for example,

described climate models as "executable theories of climate". Numerical models in Earth and environmental science embody executable theory for many different aspects of the natural world (Fig. 1). At the same time, the numerical algorithms and software that implements them provide a kind of mind-enhancing machinery. Whereas other scientific technology extends our senses—allowing us to "see" what lies beyond the visible spectrum, and to "feel" the vibrations in the Earth—computational modeling extends our cognitive capacity. By turning ideas into algorithms, we gain the ability to explore the logical conse-

quences of our ideas, make predictions, and compare them with observations. Discovery comes not only when the calculations provide self-consistent explanations for otherwise mysterious phenomena, but especially when the calculations surprise us, revealing a logic trail that leads to new insights (Bras et al., 2003).

With the rapid growth in computing and digital infrastructure, many scientists now devote a large fraction of their research time to developing software (Hannay et al., 2009; Prabhu et al., 2011; Wilson et al., 2014; Singh Chawla, 2016; Pinto et al.,

2018). A survey of nearly 2000 researchers in 40 countries by Hannay et al. (2009) revealed that 84% of respondents considered software development important for their research. According to their findings and those of Prabhu et al. (2011) scientists spend as much as a third of their time writing and debugging computer programs. In the geosciences software has become critical research infrastructure: as vital and worthy of maintenance as ships, telescopes, and seismographic arrays. Yet the invisibility of software has led to challenges in developing and sustaining this critical research infrastructure (Eghbal, 2016).

## 2.2   Growing pains

Experimental science absolutely depends on having high-quality laboratory infrastructure, and operating it with careful, systematic protocols. In this respect, computational science differs only in the invisibility of its primary infrastructure. Experimental research methods, with their emphasis on transparency and replicability, pre-date computational science by over 200 years (Wilson, 2006; Fomel and Claerbout, 2009), and so it comes as no surprise that computational science has experienced

growing pains. Errors in software can have serious consequences for research. Software faults led to the failure of the Arianne rocket in 1996, and of the Mars Climate Orbiter mission in 1999. In 2006, discovery of a bug in image-processing software led to the retraction of five papers in computational biochemistry (Miller, 2006). High-profile cases like these have sparked concern about the quality and reliability of research software. Studies of scientific software development practices underscore these concerns, suggesting that the practice of formal testing of code correctness remains relatively limited (Post and Votta,

2005; Wilson, 2006; Hannay et al., 2009; Nguyen-Hoan et al., 2010; Clune and Rood, 2011; Howison and Herbsleb, 2011; Prabhu et al., 2011; Kanewala and Bieman, 2014; Heaton and Carver, 2015). Hatton (1997) evaluated the performance of a collection of seismic data processing programs, and found that the results varied even among programs that claimed to use the same algorithm. Seeing little evidence of progress ten years later, Hatton (2007) wondered whether the scientific community must "continue building scientific castles on software sands when we could do so much better?"



Serious flaws in scientific software are not inevitable, however. Pipitone and Easterbrook (2012) found, for example, that climate models, which are subject to rigorous testing and quality controls, have very low defect density as compared with other open-source software of similar scale. Their findings show that software quality control practices can work well when applied to research products. So why are such practices not used more widely? One common obstacle is simply a lack of awareness of, and training in, effective quality-control practices such as unit testing and continuous integration (Wilson, 2006; Faulk et al.,

2009; Hannay et al., 2009; Kanewala and Bieman, 2014), a finding that led Faulk et al. (2009) to remark that "scientists are trained to manage threats to validity in experimental design but not in their codes".

A related challenge lies in computational reproducibility: the ability to recreate the results of a study using the same data and software. The ability to reproduce others' findings forms a cornerstone of the scientific method. Yet as computational science has bloomed, concern has grown over the difficulty or impossibility of reproducing published results (e.g., Schwab

et al., 2000; Peng, 2011; Stodden et al., 2013; Barba, 2016; AlNoamany and Borghi, 2018; Chen et al., 2019; Krafczyk et al., 2019). In the words of LeVeque (2009), "scientific and mathematical journals are filled with pretty pictures of computational experiments that the reader has no hope of repeating". In a reproducibility study of 306 articles in the Journal of Computational Physics, Stodden et al. (2018) found only six that provided enough method information to re-run the analysis without help from the original authors. Of the remaining papers, about half were impossible to reproduce even after contacting the authors

for assistance.

Reproducibility has several dimensions: sharing (the digital artifacts need to be available), discoverability (one needs to be able to find them), learnability (there needs to be sufficient documentation), and operability (the operating interface needs to be familiar, and the correct compute environment and dependencies must be available). Failure in any of these dimensions hurts productivity, because researchers end up spending more time either figuring out opaque, poorly documented software,

or reinventing their own version from scratch. Collectively, reports of unreproducible results and unsustainable, under-tested software suggest that computational science relies on a brittle cyberinfrastructure, and productivity suffers as a result (Wilson, 2006; Faulk et al., 2009; Prabhu et al., 2011).

A variety of factors contribute to the challenges of research software quality, reproducibility, and reusability. Most scientists lack formal training in software development, and tend not to know about tools and practices that could increase their

productivity (Kelly, 2007; Basili et al., 2008; Faulk et al., 2009; Hannay et al., 2009; Hwang et al., 2017; AlNoamany and Borghi, 2018; Pinto et al., 2018; Kellogg et al., 2018). Incentives also play a role: the academic system rewards publication of new results rather than production of high-quality, reusable software (though credit mechanisms for software are now starting to emerge) (LeVeque, 2009; Howison and Herbsleb, 2011; Morin et al., 2012; Turk, 2013; Ahalt et al., 2014; Poisot, 2015; Hwang et al., 2017; Wiese et al., 2019). The combination of incentive structure and lack of training in best practices can lead

to inflexible, hard-to-maintain software (Brown et al., 2014; Johanson and Hasselbring, 2018). Often enough it ends up as "abandonware" when a project ends (Barnes, 2010). Reluctance by code authors to provide *pro bono* support also plays a role. A certain embarrassment factor may contribute: in our own experience, as well as reports from other fields, researchers often express reluctance to share "messy" code, even when they have used the software as the basis for published research (Barnes, 2010; Morin et al., 2012; LeVeque, 2013).



## 2.3 New community practices


Despite the growing pains, there are solutions on the horizon. Tools and practices already exist that can improve the quality and efficiency of software cyberinfrastructure, and improve productivity through coordination and reuse. Practices, tools, and techniques that the software community uses routinely have begun to see uptake in the sciences, with good success (Bangerth and Heister, 2013; Turk, 2013; Hastings et al., 2014; Wilson et al., 2014; Brown et al., 2014; Poisot, 2015; Hwang et al., 2017;

Nanthaamornphong and Carver, 2017; Scott, 2017; Taschuk and Wilson, 2017; Wilson et al., 2017; Benureau and Rougier, 2018; Bryan, 2018; Adorf et al., 2018; Lathrop et al., 2019); in Section 3, we describe how the CSDMS community has implemented some of these. And while there remains a critical need for teaching and training in scientific computing, some universities, as well as community organizations such as Software Carpentry and various domain-centered groups (including CSDMS) have begun to fill that niche (e.g., Jacobs et al., 2016).

One promising development is the emergence of software journals, which provides a mean to reward research software with the academic credit it deserves. For example, the Journal of Open Source Software (JOSS), which began publishing in May 2016, focuses not on papers about results obtained by software, but instead on the "full set of software artifacts" (Smith et al., 2018). Reviewers of JOSS submissions evaluate the software directly; a one or two page abstract describing the purpose and function of package forms the only textual component, apart from documentation. For the Earth and environmental sciences,

JOSS now complements more traditional text-based journals, such as Geoscientific Model Development, that provide a forum for software-oriented issues such as algorithm development and model verification. The growing importance of software in research has also led to a new type of career track: Research Software Engineers (RSEs), whose cross-training in computing and domain science positions them to help researchers build and maintain high-quality, sustainable software (Baxter et al., 2012). Thus, the academic world now has the beginnings of a credit mechanism that incentivizes high-quality research software

cyberinfrastructure, and the first glimmers of a professional structure to help create and maintain that cyberinfrastructure.

Better incentives and support for writing, documenting, and publishing research software can help address the productivity problem because they encourage software reuse over reinvention. Community software libraries and modular frameworks provide another avenue for reuse. Libraries are already widely available for general tasks such as numerical computing, parallel programming, and general science and engineering operations; some examples include PETSc (Abhyankar et al., 2018), Deal.II

(Bangerth et al., 2007), and the SciPy family (Virtanen et al., 2020). "Librarization" of software makes it easier to share, reuse, and maintain (Brown et al., 2014). Frameworks are defined as a collection of interoperable modules together with an environment for running and combining them. A framework provides a way to create coupled numerical models, and more generally to simplify computational workflows (e.g., Leavesley et al., 1996; Voinov et al., 2004; Peckham et al., 2013). Frameworks, as well as some open-source libraries, take advantage of contributions from many different community members:

the software becomes a resource created by and for a scientific community. Growth of a community framework does not happen by accident, however. Case studies of community frameworks, libraries, and other software packages reveal that success requires two elements: a thoughtful, deliberate approach to community engagement (Bangerth and Heister, 2013; Turk, 2013; Lawrence et al., 2015), and carefully designed standards and protocols (Peckham et al., 2013; Harpham et al., 2019).





## 3 A community-based modeling system for Earth-surface processes

The opportunities and growing pains that face scientific computing generally also apply to the sciences that deal with the Earth's surface. To embrace these opportunities, the CSDMS Integration Facility was launched in 2007 with a mission to accelerate the pace of discovery in Earth-surface processes research. The centerpiece was envisioned as "a modeling environment containing a community-built, freely available suite of integrated, ever-improving software modules aimed at predicting the erosion, transport, and accumulation of sediment and solutes in landscapes and sedimentary basins over a broad range of time and space scales" (Anderson et al., 2004). A key concept is that a modular, community-built modeling system not only opens new opportunities for using coupled models to explore the interactions among processes that were once considered in isolation, but also increases productivity by lowering the kinds of barriers described earlier. Achieving this requires a combination of:

– **Community**: coordination, sharing, communication, collaboration (e.g., conferences, workshops, hackathons);

– **Computing**: software tools, standards, templates, access to high-performance computing and cloud resources;

– **Education**: in-person and online resources for learning tools, techniques, and best practices; resources for teaching these to others.

In the following sections, we describe the software technology, community building, and education elements developed by CSDMS, and how they help mitigate the obstacles discussed in Section 2. A useful way to understand the purpose of these products and activities is to consider the different modes in which researchers operate numerical models, and the opportunities that these different modes present to increase efficiency and productivity.

## 4 A taxonomy of model operation

What tasks are often required in computational modeling? And how might those tasks be made efficient? Here we identify six types of model-related activity, each of which has a unique set of challenges. Inspired by Bloom's Taxonomy of cognitive learning tasks, these six activities are arranged in order of complexity. The six modeling modes are summarized in Fig. 2.

### 4.1 Reproducing

The most basic operation of a numerical model is to run it with predefined inputs. This is often the first step in learning how to use a particular model. The ability to reproduce a model calculation efficiently involves all four of the FAIR principles (Wilkinson et al., 2016). The user must be able to find and access the right version of the software. The user needs to learn how to execute the model: a task made easier if the program follows an interoperability standard. In order to reproduce the prior calculation, the user must have access to the input data, and must be able to recreate a compatible execution environment, including whatever dependencies might be needed.





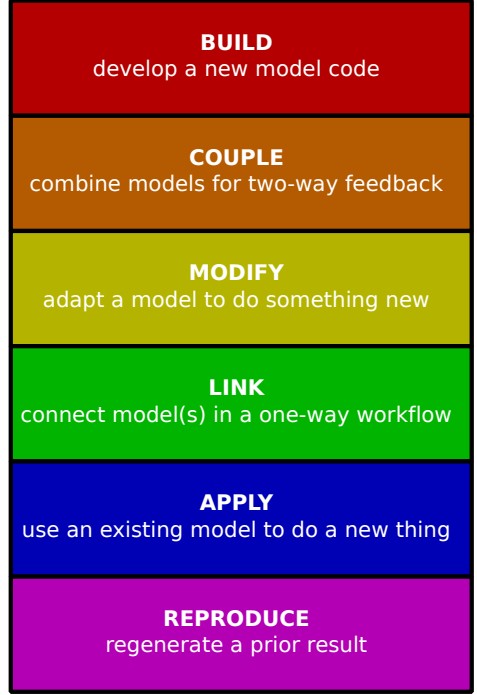

**Figure 2.** Taxonomy of model operation tasks.

## 4.2 Applying

To use a computational model in a new application, a user needs to understand the theory and algorithms behind it, and that requires good documentation. In addition, operating the model involves creating new input data, and often executing various

pre-processing operations to derive the right kind of inputs. Sometimes it also requires setting up a grid mesh. In some cases, mesh generation is a major undertaking, for example, meshes for 2D storm-surge models such as ADCIRC (Luettich et al., 1992) and 3D regional ocean circulation models such as ROMS (Shchepetkin and McWilliams, 2005) are time-intensive to set up.

## 4.3 Linking

Here *linking* means operating a model as part of a sequential workflow. For example, the workflow might include pre-processing data, using those data as input to the execution of a model, and using the output as input to another model, and/or performing additional operations on the model's output. To link a model in this way requires, among other things, compatibility in data formats. Any incompatibility between the outputs from one step and the inputs to the next means someone has to write code to do the appropriate translation.





## 4.4 Modifying

When a model provides most but not all of the functionality needed for a particular application, the would-be user faces a choice: modify the existing program, or write a new one that fits the purpose. Modifying an existing model program can save a lot of duplication of effort, but only when the model package includes good internal documentation, a modular design, and a structure that allows for modifications and enhancements while preserving the original functionality. A standard interface design can help by providing a familiar structure.

## 4.5 Coupling

Many of the exciting research frontiers in Earth and environmental science lie at the seams between systems. Some examples include: rivers and coasts (e.g., Ratliff et al., 2018); tectonics and Earth-surface processes (e.g., Roy et al., 2016); ecosystems, soils, and landscape evolution (e.g., Istanbulluoglu and Bras, 2005; Pelletier et al., 2017; Lyons et al., 2020); permafrost and hydrology, and human actions and biophysical systems (e.g., Robinson et al., 2018). For these sorts of problem, coupled numerical modeling provides a great way to develop insight and to test hypotheses by comparing models with observations. The complexity of the task of coupling two numerical models depends on the nature of coupling (for example, sequential execution within each time step, versus coupling via joint matrix inversion) and on the program structure of each. The task becomes much simpler when both models offer a public, standardized interface: a set of callable functions that allow the appropriate exchange of data and mutual execution of algorithms.

## 4.6 Building

New ideas stimulate the need for new models. It is a healthy sign of growth when a scientific community produces lots of new models, because it signifies rapid development and exploration of new concepts. Writing a numerical model program from scratch can be a time-consuming exercise. Libraries of pre-existing functions and data structures can greatly simplify the task. Most modern programming languages offer libraries to handle basic mathematical operations, but even with these available, model-building can be a major effort.

The job becomes easier when the developer can draw on component libraries that provide data structures and algorithms to address common tasks in numerical modeling, such as grid setup and input/output. It becomes easier still when common domain-specific algorithms have been librarized and made available as building blocks with a standard interface (e.g., Brown et al., 2014). Below we will look at an example of a component library that was designed specifically for building numerical models.

## 5 The CSDMS model repository: a platform for sharing and archiving software resources

Not so long ago, making model source code freely available was more the exception than common practice. Model developers tended to view their models as trade secrets. If others wanted to use a model, the developer needed to be contacted, and could





negotiate to become more involved in the research. Furthermore, fewer tools and platforms were available to promote sharing, like GitHub (established 2008) or SourceForge (established 1999).

Science clearly benefits from openly shared source code. For one, sharing reduces duplication. After all, there is less need to write a model from scratch, once a model has proven to capture a certain process well. Therefore, sharing of source code accelerates science, as others are on a faster trajectory to learn from and build upon previous model development efforts. Shar-
240 ing of source code also makes science more robust and trusted, as people can report and fix bugs. Reproducing computational results requires shared source code. It is therefore encouraging to see a modeling culture shift over the last two decades (e.g., Hsu et al., 2015). For good data management, there are now the FAIR principals —Findability, Accessibility, Interoperability, and Reusability—that have been formulated as guidelines for data producers and publishers (Wilkinson et al., 2016). According to the FAIR principles, each dataset should have a unique digital object identifier (DOI) to get to the data, associated
with searchable metadata. By including a formal, broadly applicable representation language, and using open and widely accepted domain-relevant vocabularies and ontologies, data sets become more interoperable. And by providing an abundance of documents that describe datasets and how they can be used, including license information, data become more reusable.

For model code, version-control platforms are now more widely used for sharing source code. But as the FAIR-data principles indicate, sharing code by itself is not enough. Therefore, CSDMS implemented the FAIR principles in setting up a model
repository for Earth surface dynamics (Fig. 3). A minimal set of metadata parameters is defined to: describe a model, provide contact information for the model development team, indicate technical details such as operating platform and the software license, describe the model input and output, list its processes and key physical parameters, and indicate limitations. This minimal set of metadata includes a link to the actual source code, which needs to be made available 24/7 through a personal web repository, or through the CSDMS community repository. All model metadata stored on the CSDMS web server, as well as the
actual source code (when stored in the CSDMS code repository on GitHub) are accessible for machines through web APIs. This makes it possible to automatically find and use the model. DOIs for stable versions of any listed code are generated on request and included with the metadata. Model metadata are enriched by including additional reference information, such a comprehensive bibliography. Following this practice, the CSDMS model repository currently holds 387 open source models of the community (as of Feb 2021). The models and tools in the repository span a range of languages, with Python, C, and
Fortran being the most popular (Fig. 4). The diversity of languages raises a challenge in creating an interoperable framework. We will return to this point, and look at one solution, in Section 6.2.

# 6 The CSDMS Workbench

The **CSDMS Workbench** is a suite of tools, standards, documents, and resources that collectively provide a modular environment for model execution, analysis, and model-data integration. The Workbench comprises six main elements:

1. The **Basic Model Interface (BMI)**: an interface standard that simplifies model execution and coupling (Hutton et al., 2020; Peckham et al., 2013).

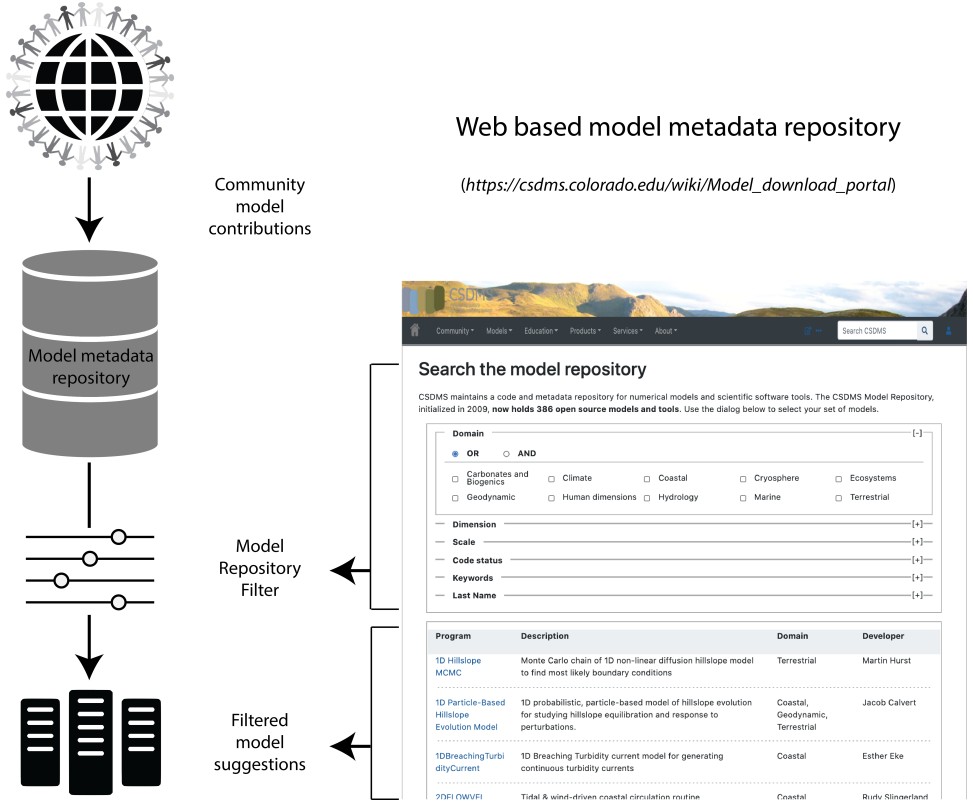

**Figure 3.** The CSDMS Model Repository.

2. **Babelizer**: a language-bridging tool that adds a Python interface to BMI-enabled model programs written in various other languages.

3. **pymt**: a Python-language execution and model-coupling environment, which includes utilities for grid mapping and other operations, together with a set of model components.

4. **Data Components**: small Python-language modules that use the BMI to fetch data from particular data sets.

5. **Landlab**: a component-based Python library for model building and sharing of interoperable components (Hobley et al., 2017; Barnhart et al., 2020a).

6. **Standard Names**: An ontology standard for naming variables.

In the following we give a brief description of each of these elements, and how they combine to form a modular modeling system.

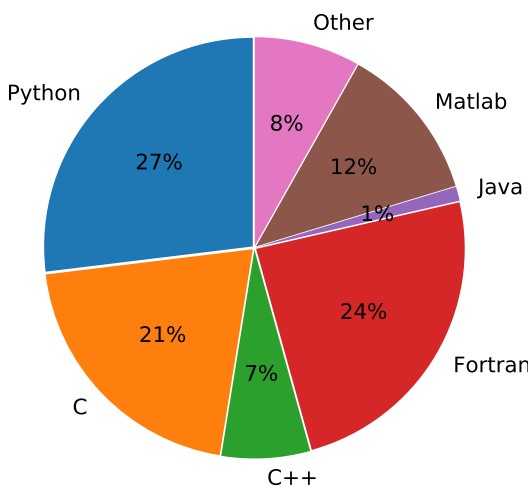

**Figure 4.** Distribution of different languages used by models and tools in the CSDMS Model Repository (percentages out of 453 total programs; not weighted by program size).

## 6.1 The Basic Model Interface (BMI) Standard

When you sit in the driver's seat of an unfamiliar car, you're presented with a familiar sight: whatever the make or model, the vehicle provides a steering wheel, brake pedal, and speedometer. Although we don't usually think of it this way, drivers across the globe benefit from a *standard interface*—a set of control mechanisms and information displays that have essentially the same design regardless of whether the car is a tiny electric two-seater or a giant stretch limousine. This standard interface makes operating a car much easier than if each vehicle presented a radically different interface. Imagine a world where switching from a sports car to a pickup truck required months of study and practice!

We believe numerical models should offer a similar standardization. To this end, CSDMS developed the Basic Model Interface (BMI) (Peckham et al., 2013; Hutton et al., 2020). In software engineering, an interface is a named set of functions with prescribed arguments and return values. The BMI provides a standard set of functions for querying and controlling a model. Just as with a car, when a model is equipped with a BMI, it becomes easier to use because its control functions are now the same as every other model with a BMI.

Further, because BMI includes variable-exchange functions, a model with a BMI can be coupled with other models that expose a BMI. Tables 1 and 2 list the individual functions that comprise the Basic Model Interface, along with a brief description of each. The table shows the current version of BMI, version 2.0, which represents a collection of improvements to the original specification, especially in the representation of model grids (Hutton et al., 2020). A model program that has been *wrapped* with a BMI can function as an interoperable *component*, which can be combined with others to create integrated models (Fig. 6).





**Table 1.** Listing and description of Basic Model Interface (BMI) functions for run control and data access.

| Function Name | Description |
| --- | --- |
| `initialize` | Perform startup tasks for the model. |
| `update` | Advance model state by one time step. |
| `update_until` | Advance model state until the given time. |
| `finalize` | Perform tear-down tasks for the model. |
| `get_component_name` | Name of the model. |
| `get_input_item_count` | Count of a model's input variables. |
| `get_output_item_count` | Count of a model's output variables. |
| `get_input_var_names` | List of a model's input variables. |
| `get_output_var_names` | List of a model's output variables. |
| `get_var_grid` | Get the grid identifier for a variable. |
| `get_var_type` | Get the data type of a variable. |
| `get_var_units` | Get the units of a variable. |
| `get_var_itemsize` | Get the size (in bytes) of one element of a variable. |
| `get_var_nbytes` | Get the total size (in bytes) of a variable. |
| `get_var_location` | Get the grid element type of a variable. |
| `get_current_time` | Current time of the model. |
| `get_start_time` | Start time of the model. |
| `get_end_time` | End time of the model. |
| `get_time_units` | Time units used in the model. |
| `get_time_step` | Time step used in the model. |
| `get_value` | Get a copy of values of a given variable. |
| `get_value_ptr` | Get a reference to the values of a given variable. |
| `get_value_at_indices` | Get variable values at specific locations. |
| `set_value` | Set the values of a given variable. |
| `set_value_at_indices` | Set the values of a variable at specific locations. |





**Table 2.** Listing and description of Basic Model Interface (BMI) functions for querying grid data.

| Function Name | Description |
| --- | --- |
| `get_grid_rank` | Get the number of dimensions of a computational grid. |
| `get_grid_size` | Get the total number of elements of a computational grid. |
| `get_grid_type` | Get the grid type as a string. |
| `get_grid_shape` | Get the dimensions of a computational grid. |
| `get_grid_spacing` | Get the spacing between grid nodes. |
| `get_grid_origin` | Get the origin of a grid. |
| `get_grid_x` | Get the locations of a grid's nodes in dimension 1. |
| `get_grid_y` | Get the locations of a grid's nodes in dimension 2. |
| `get_grid_z` | Get the locations of a grid's nodes in dimension 3. |
| `get_grid_node_count` | Get the number of nodes in the grid. |
| `get_grid_edge_count` | Get the number of edges in the grid. |
| `get_grid_face_count` | Get the number of faces in the grid. |
| `get_grid_edge_nodes` | Get the edge-node connectivity. |
| `get_grid_face_edges` | Get the face-edge connectivity. |
| `get_grid_face_nodes` | Get the face-node connectivity. |
| `get_grid_nodes_per_face` | Get the number of nodes for each face. |

While a BMI can be written for any language, CSDMS currently supports four languages: C, C++, Fortran, and Python. A simple example of using a BMI written in Fortran is shown in Listing 1 below. The model shown in this example is the surface

**Listing 1** Example Fortran BMI code.

```
type (bmi_prms_surface) :: model
character (len=*) :: config_file
integer :: status
double precision :: now, end_time

status = model%initialize(config_file)
status = model%get_current_time(now)
status = model%get_end_time(end_time)
do while (now < end_time)
   status = model%update()
   status = model%get_current_time(now)
end do
status = model%finalize()
```





water component of the Precipitation Runoff Modeling System (PRMS), developed by the U.S. Geological Survey (Leavesley et al., 1983). In the example, the model is initialized from its native configuration file, then stepped forward in time until it reaches its stop time, whereupon any resources it uses are deallocated. Note that only BMI function calls are used to drive the model; no knowledge of the underlying calls to control PRMS is needed.

Hoch et al. (2019) provide a current research example of using BMI. In their study, they coupled a hydrologic model, PCR-
GLOBWB, with a pair of hydrodynamic models, CaMa-Flood and LISFLOOD-FP, through BMI. They observed that a coupled model system enhanced the accuracy of peak discharge simulations (Fig. 5). Hoch et al. conclude that "results confirm that

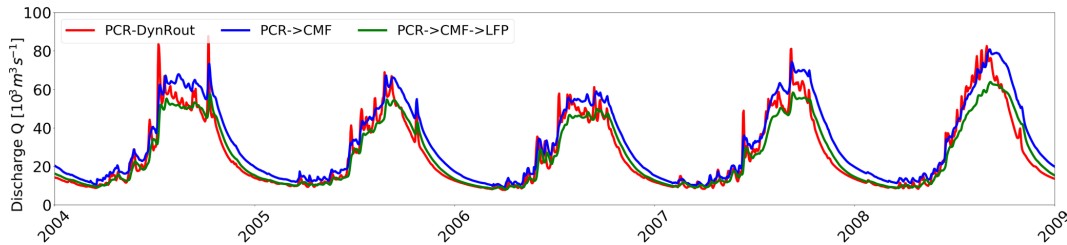

**Figure 5.** Simulated discharge at a river monitoring station from coupled hydrologic and hydrodynamic models. (Reproduced from Figure 6 in Hoch et al. (2019).)

model coupling can indeed be a viable way forward towards more integrated flood simulations. However, results also suggest that the accuracy of coupled models still largely depends on the model forcing."

### 6.2  Language Interoperability: The Babelizer

Looking at Fig. 4, we notice that software generated by the CSDMS community reflects a range of programming languages and, so, language interoperability is critical to a coupling framework if it is to bring together this diverse set of models.

One approach to solving this problem is to choose a *hub* language through which other languages will communicate. An advantage of this approach is that it needs only to provide bridges from each supported language to the hub language, rather than building bridges for each language to every other language. CSDMS uses Python as a hub language for several reasons: it
is open source, has a large user base in the user community, has an active community that supports a vast library of 3rd-party packages (numpy, scipy, xarray, pandas, etc.), and, importantly, there are existing pathways to bring many other languages into Python.

The *babelizer* is a command-line utility CSDMS created to streamline the process of bringing a BMI component into Python. For libraries that expose a BMI, the *babelizer* creates the necessary glue code to create a Python importable package
that presents the BMI component as a Python class. We wrote the *babelizer* to be easily extensible to additional languages but, presently, it can be used to wrap libraries written in C, C++, and Fortran using the Cython language.





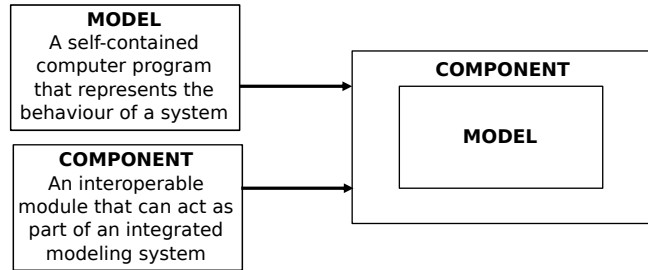

**Figure 6.** When a numerical model program is "wrapped" with a standard interface and compiled as an importable module, it becomes an interoperable *component*.

## 6.3 Execution and Coupling Framework: pymt

Models in the Earth Sciences are as diverse as the environments they are intended to represent. Codes are written by hundreds of authors, in different languages, from a diverse range of domains, operate on time and space scales that span orders of magnitude, and are oftentimes written in isolation—never intending to be used by someone outside the core development team. And so, models do not fit so neatly together.

Although the CSDMS collection of models is incredibly diverse, there is a common thread possible—the Basic Model Interface (BMI; Section 6.1) —that connects them and allows us to create tools that allow scientists to easily pick up, run, and even couple these models with one another. While only a subset of codes in the Model Repository (Section 5) provide a BMI, the concept is general enough that any model can be given one. To provide a framework for operating and coupling these BMI-equipped codes, the CSDMS Integration Facility develops and maintains a Python package known as the *pymt* (Python Modeling Toolkit).

The CSDMS Integration Facility has written the pymt as a Python package that gives scientists a set of utilities for running and coupling Earth System models within a Python environment. We see the pymt as, primarily, two things: (1) a collection of Earth-surface models, in which every model exposes a standardized interface (and so, if you are able to run one model, you will be able to run any model), and (2) tools needed for coupling models across disparate time and space scales. A key feature of pymt is extensibility: any contributor can implement a BMI and use the babelizer (6.2) to create to add a new model or utility to the toolkit.

Although the pymt itself is written in Python, the models in its collection need not be written in Python. The babelizer allows developers and contributors to bring models from other languages into a Python environment. The current pymt model collection is detailed in Table 3. One thing to note when reading through this list, apart from the diversity of models, is that models span a range of granularity (that is, the size of a model's scope). Granularity ranges from a single equation (for example, from hydrology, the Richards equation or Green-Ampt method to model infiltration) to a collection of coupled process models (or even a complete modeling framework; e.g. *CHILD* (Tucker et al., 2001) or *Sedflux3D* (Hutton and Syvitski, 2008)). However, we find that the most useful model size is one between these two extremes, which simulates a single physical process





**Table 3.** Models available as components in the Python Modeling Tool

| Component | Property / Process(es) | Language |
|---|---|---|
| FrostNumber | permafrost occurrence | Python |
| Kudryavtsev (Ku) | soil active layer thickness | Python |
| GIPL | soil temperature & heat flow | Fortran |
| ECSimpleSnow | snow balance | Fortran |
| HydroTrend | river water & sediment discharge | C |
| RAFEM | river avulsion & floodplain evolution | Python |
| CHILD | landscape evolution | C++ |
| CEM | sandy, wave-dominated coastline evolution | C |
| GridMet | meteorological data access | Python |
| Sedflux3D | seafloor evolution | C |
| Avulsion | river avulsion | C |
| Plume | hypopycnal sediment plume | C |
| Subside | 2D lithospheric flexure | C |
| OverlandFlow* | Surface water runoff | Python |
| Flexure* | 1D/2D lithospheric flexure | Python |
| LinearDiffuser* | 2D linear diffusion | Python |
| ExponentialWeatherer* | Weathering of bedrock on hillslopes | Python |
| TransportLengthHillslopeDiffuser* | Hillslope diffusion | Python |
| Vegetation* | productivity, biomass and leaf area index | Python |
| SoilMoisture* | Root-zone soil moisture | Python |
| DepthDependentDiffuser* | Depth and slope dependent linear diffusion | Python |
| FaSTMECH | River flow and morphodynamics solver | Fortran |
| PRMSStreamflow | River channel flow (Muskingum routing) | Fortran |
| PRMSGroundwater | Subsurface water flow | Fortran |
| PRMSSoil | Soil zone flow | Fortran |
| PRMSSurface | Surface water flow | Fortran |

(* = Landlab component)





(for example the compaction of sediment under an overlying load, or the transport of sediment by way of hypopycnal sediment plumes). Models of this size are flexible in the number of other models they can couple with but not so small that they do not justify the extra overhead of creating a separate component.

We have included with the pymt a collection of tools a modeler can use to connect a disparate set of models. For example,
models will not necessarily operate on the same spatial grid and so may have different spatial resolutions or even different grid types (e.g., raster versus unstructured mesh). To overcome this problem, we use the Earth System Modeling Framework (ESMF) grid mapper, which uses interpolation to translate variables between grids. Using this grid mapper, a modeler can write a script that gets grid values from one model and pymt will automatically map them onto the grid of another (Fig. 7).

Another common issue when exchanging data between models in a coupled is system is unit mismatches. To address this
issue, the pymt contains a Python wrapped version of the *udunits* unit conversion library. When connecting components within pymt, the user specifies the units for quantities that components will either use or provide. As with grid mapping, the pymt decorates the standard BMI with additional functionality, thus leaving these common tasks to the framework rather that to the developer of each model. The two quantity converters (grid mapping and unit conversion) target the BMI `get_value` methods, that is differences in quantities defined on a spatial grid. However, two models can also differ temporally.

Depending on a model's time resolution, the algorithm it uses to solve a set of equations, or the time scale being simulated, models may not advance forward time at compatible intervals. However, when coupling models, we require the exchange of quantities to be made when models are synchronized in time. While the BMI `update_until` method could be used for this, we recognize (for some of the reasons listed above) that not all models can realistically implement this method. For such cases we've added time interpolators to the pymt by way of a modified `update_until` method that estimates values at intermediate
time steps. The pymt accomplishes this by temporarily saving quantities at previous time steps and then interpolating between time steps. Consider, for example, a user who wants to couple two models: the first advances in time at $\Delta t$ and the second at a larger time step of $2\Delta t$. Both models sit at time $t_0$ but the first wants to get a quantity, $x(t)$, from the second at $t = t_0 + \Delta t$. To do so, pymt advances the second model by its time step to $t = t_0 + 2\Delta t$ and returns an interpolated value of $x(t_0 + \Delta t)$. pymt does this behind-the-scenes within the second model's `update_until` method.

Figure 7 shows the results of a coupling experiment that demonstrates some of pymt's capabilities. Here we have coupled the landscape evolution model CHILD with the seascape evolution model sedflux3D. The landscape is uplifted and eroded by CHILD, including fluvial transport of sediment to the coast. At the coast, sedflux3D takes over and transports sediment to the seafloor through surface sediment plumes and, over time, builds up a delta, which becomes part of the subaerial landscape (and thus part of the domain of CHILD). For every time step, CHILD passes river fluxes to sedflux3D which, in turn, passes updated
landscape elevations back to CHILD. Apart from the difference in domains (land vs. sea), the two models also differ in their computational grids: CHILD uses an unstructured mesh while sedflux3D uses a uniform rectilinear grid. The pymt manages the time stepping, variable exchange, and the mapping of variables between the two grids.

In addition to providing a set of coupling tools, pymt provides an interactive environment to couple and run models. Although the two models in our previous example were written in C (sedflux3D) and C++ (CHILD), when imported as Python classes
in pymt, users are able to instantiate and run the two models interactively. A user advances models one time step at a time and

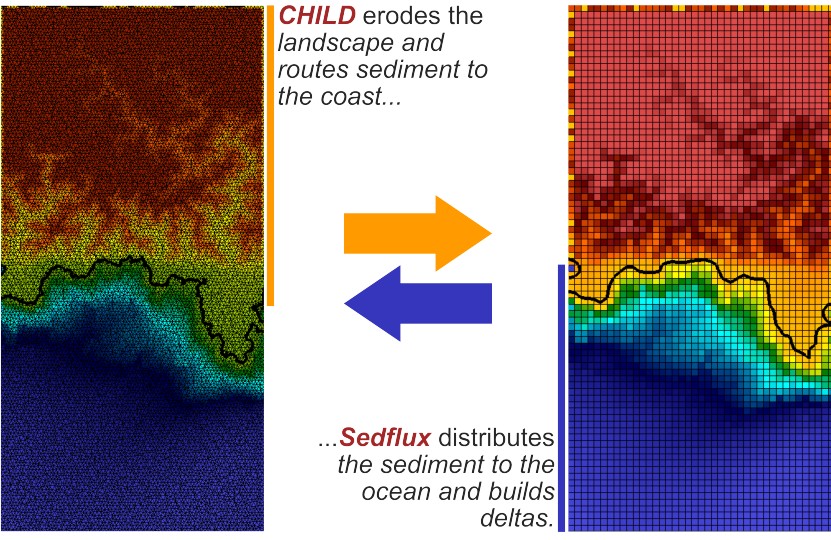

**Figure 7.** Results of a coupling experiment using the Python Modeling Toolkit (pymt). The CHILD model (C++; unstructured mesh) erodes an uplifting landscape and transports sediment to the coast. Sedflux (C; structured rectilinear grid) sends sediment from the coast to the ocean as surface plumes where it then settles onto the seafloor. As sediment accumulates on the seafloor, deltas begin to form and this new landscape is sent back to CHILD, completing the two-way coupling between these two pymt components.

can then query or even change values dynamically. When run in their native languages, a user would set the initial conditions for a model simulation and then let the model run to completion before examining output. A user would never be able to dynamically change values as it advanced. The functionality that pymt provides allows a user to experiment interactively, examining state variables as the model evolves, and dynamically changing model state variables as it advances—all within a

Python environment with its large collection of visualization and analysis packages.

### 6.4 Data Components

Researchers rarely use numerical models in isolation. Working with models nearly always includes working with data sets too: the data that go into a model as input, the output data that a model produces, and the data to which a model's output is compared (Fig. 2). Productivity suffers when these data sets are cumbersome to access and use. Just as model interface standards like

BMI make it easier to work with numerical models, standardized methods for data access and retrieval can ease the burden of working with data. To that end, CSDMS has developed a programmatic approach that uses the BMI for data retrieval and access. Functions such as `initialize()` retrieve and open a data set, and `get_value()` fetches particular data items or subsets. A program that uses the BMI to access items from a particular data set is known as a *Data Component*. Using the same interface for model and data operation makes it easier to swap models and data sets; for example, one might compare



use of model-calculated versus measured wave heights in a simulation of coastal sediment transport. Because CSDMS Data
Components are written in Python, they can take advantage of data-management packages like `pandas` and `xarray`.

The data components are designed to provide a consistent way to access various types of datasets (e.g., time series, raster
grid, and multidimensional space-time data) and subsets of them without needing to know the original file formats. Each
data component effectively "wraps" a dataset with a BMI (with the exception of certain BMI functions, such as `set_value`,
which do not apply to datasets). Data components can easily interact with BMI-enabled numerical models in the pymt modeling
framework, or other similar frameworks.

One example is the National Water Model (NWM) data component. This data component can access and subset the fore-
casted streamflow time series generated by the NWM hydrologic modelling framework. Figure 8 shows an example of how the
NWM data component can be used to get the streamflow data at a river channel for a flooding event. Figure 9 shows the corre-
sponding time series plot. This data component includes a set of standard control and query functions (e.g., `initialize()`,
`update()`). These standard methods make the dataset easier to couple with BMI-enabled numerical models without needing
to know the time-series file format.

## 6.5 Creating new models: Landlab

Landlab is a Python-language library designed to support the creation, combination, and re-use of 2-D models (Hobley et al.,
2017; Barnhart et al., 2020a). For the moment, let's presume that a model developer has identified input and output parameters,
model state variables, and the governing equations and/or model rules. We might then synthesize the tasks of building the model
(Section 4.6) into two types: (a) creating required data structures, and (b) implementing a numerical solution to the governing
equations that act on those data structures. For example, most models need to represent the computational domain, including
information across the domain, and adjacency information describing how the different parts of the domain are connected
to one another. This division is simplistic, and neglects many intricacies, yet it captures the fundamental activities of model
building.

Landlab provides re-usable software infrastructure that addresses the most common needs for our two model-building tasks.
For grid-based data structures, Landlab provides a *grid* object to represent the computational domain and store *fields* of state
variables (Fig. 10). Landlab provides several two-dimensional grid types, which all share the same underlying graph-based
data structures. Current grid types include regular raster, network, regular hexagon, and unstructured (Delaunay/Voronoi). For
all grid types, the adjacency information and access to fields follows the same interface—making it easier for a model to work
on multiple grid types.

To address the second model-building task, Landlab provides two capabilities. First is a set of numerical utilities that support
common needs. These include, for example, the ability to calculate differences, gradients, fluxes, and divergences of values
stored at fields. Second is a library of *Components* (Fig. 11). Each Landlab component simulates a single process, such as
routing of shallow water flow across a terrain surface (Adams et al., 2017), calculating groundwater flow (Litwin et al., 2020),
modeling sediment movement in a river network (Pfeiffer et al., 2020), or simulating biological evolution across a landscape
(Lyons et al., 2020). Components are implemented as Python classes, and are derived from a common base class that defines





**Example: using the National Water Model (NWM) Data Component to download data**

The NWM Data Component is implemented with a Python class called **BmiNwmHs**.

Import BmiNwmHs class and instantiate it. A configuration file (yaml file) is required to provide the parameter settings for data download.

```
In [1]: import matplotlib.pyplot as plt
        import numpy as np
        import cftime

        from nwm import BmiNwmHs

        # initiate a data component
        data_comp = BmiNwmHs()
        data_comp.initialize('config_file.yaml')
```

Use variable-related methods from BmiNwmHs class to check the variable information of the NWM dataset. This data component stores a flow forecast variable.

```
In [2]: # get variable info
        var_name = data_comp.get_output_var_names()[0]
        var_unit = data_comp.get_var_units(var_name)
        print(' variable_name: {}\n var_unit: {}\n'.format(var_name, var_unit))

        variable_name: Flow Forecast
        var_unit: cfs
```

Use time-related methods to check the time information of the NWM dataset. The time values are stored in a format that follows the CF convention (http://cfconventions.org/Data/cf-conventions/cf-conventions-1.8/cf-conventions.pdf).

```
In [3]: # get time info
        start_time = data_comp.get_start_time()
        end_time = data_comp.get_end_time()
        time_step = data_comp.get_time_step()
        time_unit = data_comp.get_time_units()
        time_steps = int((end_time - start_time)/time_step) + 1
        print(' start_time:{}\n end_time:{}\n time_step:{}\n'
              ' time_unit:{}\n time_steps:{}\n'.format(start_time,
                                                       end_time,
                                                       time_step,
                                                       time_unit,
                                                       time_steps))

        start_time:1503450000.0
        end_time:1503511200.0
        time_step:3600.0
        time_unit:seconds since 1970-01-01 00:00:00 UTC
        time_steps:18
```

Loop through each time step to get the flow and time values. stream_array stores flow forecast values. cftime_array stores the numerical time values. time_array stores the corresponding Python datetime objects. get_value( ) method returns the flow forecast value at each time step. update( ) method updates the current time step of the data component.

```
In [4]: # initiate numpy arrays to store data
        stream_value = np.empty(1)
        stream_array = np.empty(time_steps)
        cftime_array = np.empty(time_steps)

        for i in range(0, time_steps):
            data_comp.get_value(var_name, stream_value)
            stream_array[i] = stream_value
            cftime_array[i] = data_comp.get_current_time()
            data_comp.update()

        time_array = cftime.num2date(cftime_array, time_unit, only_use_cftime_datetimes=False)
```

**Figure 8.** Screenshot of a JupyterNotebook demonstrating how to use the NWM Data Component to access and subset a streamflow dataset.

common attributes, and enforces a minimum set of metadata for each component. If a researcher wishes to write the code for

a numerical model, and the desired elements of that model have already been implemented as Landlab components, the model



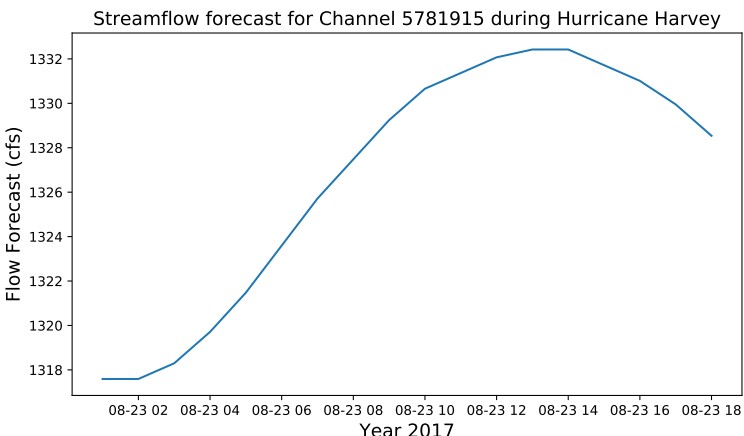

**Figure 9.** Time series plot for the accessed NWM streamflow dataset.

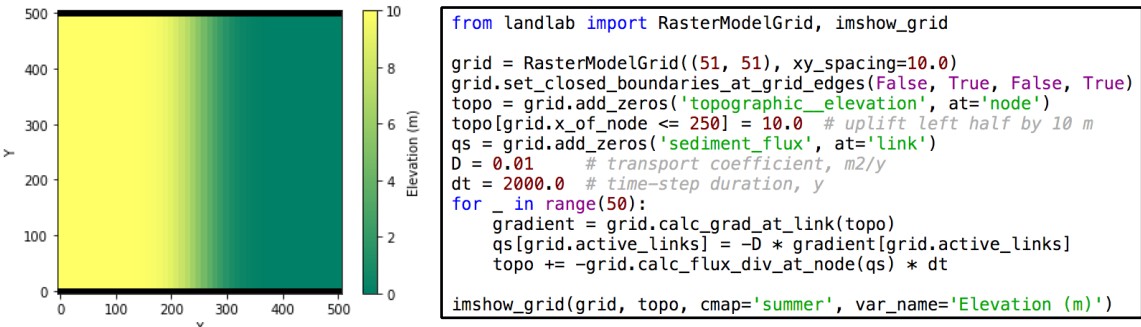

**Figure 10.** Example of a simple finite-volume numerical model of hillslope evolution, written in Python using the Landlab library. Model uses a diffusion equation to represent an evolving hillslope. The concise source code at right illustrates the use of a Landlab *grid* object together with *fields* and built-in functions for numerical gradient and divergence functions on these fields.

can be programmed efficiently by instantiating each component, and then executing the `run_one_step` method for each component within a loop (Fig. 11).

The *Component* base class was designed to expose a Basic Model Interface (BMI; Section 6.1), which allows a Landlab component to be used as a BMI-enabled component. Although we do not expect most Landlab users to directly use this

alternate interface, the component's BMI acts as a bridge that allows it to be incorporated into other BMI-friendly frameworks and tools (e.g., pymt, *dakotathon*).

The Python Modeling Toolkit (pymt; Section 6.3) is a model-coupling framework that provides tools for using and coupling BMI-enabled components, written in a range on programming languages, which may not have been written with the intent of operating within a coupling framework. Operating as a BMI component, Landlab components act as isolated elements that no



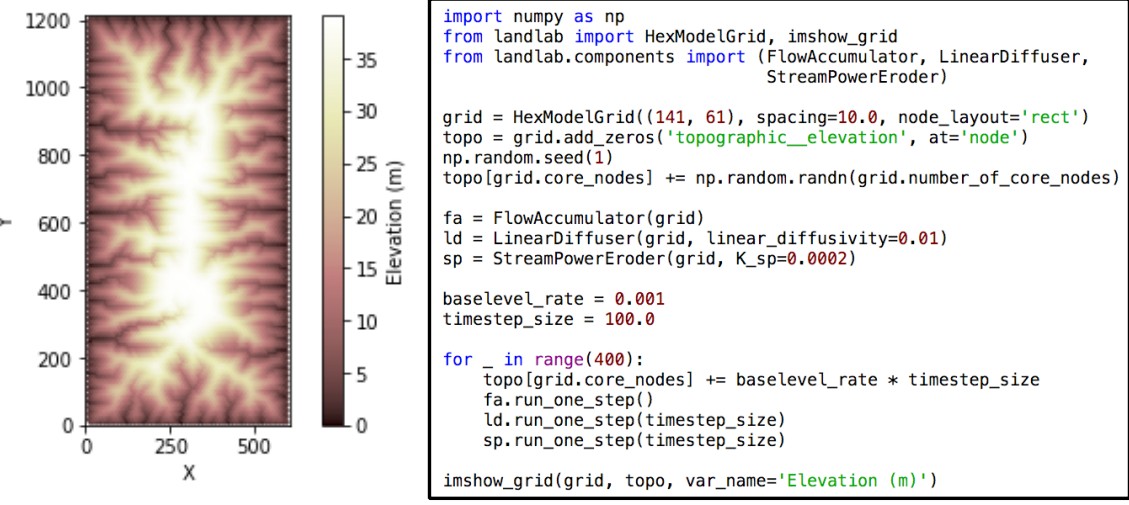

**Figure 11.** Example of a simple landscape evolution model, written using the Landlab programming library. Model code at right illustrates the use of three *components* to create the model.

longer share a common grid and data; when used in this mode, Landlab components require an input file that describes the grid, parameter values, and initialization setup. This is by design and required by the BMI so that a user interacts with the component as with any other BMI component without being aware of the inner workings of a Landlab component.

Despite the name, Landlab is not restricted to terrestrial processes. Its component collection includes, for example, components for coastal and marine processes such as tidal circulation and marine sedimentation. Its design is amenable to a wide

variety of 2D grid-based numerical models and cellular automata applications. Landlab can be used, for example, to construct integrated source-to-sink models that treat the full geologic cycle, tracking sediment from its creation on land to its deposition in marine basins (Fig. 12).

The design of Landlab supports a variety of usage styles. Interested users and/or developers may use Landlab to create models as Components, or as scripts that combine components. Alternatively, Landlab can be used to build stand-alone packages

packages such as `terrainbento` (Barnhart et al., 2019), which combine Landlab components into a predefined set of models.

### 6.5.1 HyLands: an example of a component-based integrated model

The modular design of Landlab enables the development of numerical tools in an efficient manner. An example of a recently developed Landlab-built model is HyLands: a landscape evolution model that simulates mass wasting and sediment redistri-

bution on hillslopes. The model was originally writte in a closed-source language (Campforts et al., 2020); translating the original code into Landlab converts the original product into a fully open-source tool for the broader community, and provides a new process component to simulate landsliding. The grid engine and other tools available within the Landlab library enabled efficient implementation and provide capabilities for coupling with other, existing Landlab components. An example



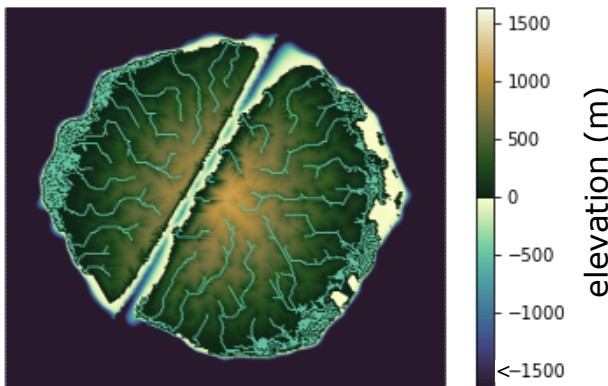

**Figure 12.** Snapshot of prototype integrated numerical model of landscape and sedimentary basin evolution. Domain size is 250 by 250 km. Simulation shows a hypothetical micro-continent with an active NNE-oriented extensional fault. Sea level varies stochastically; this particular snapshot captures a period of rising sea level after a brief low-stand. Model was constructed using Landlab components for flow routing (*FlowAccumulator, FlowDirectorSteepest, LakeMapperBarnes*), fluvial processes (*ErosionDeposition*), marine sediment transport (*SimpleSubmarineDiffuser*), lithosphere flexure in response to loading/unloading (*Flexure*), and extensional faulting (*ListricKinematicExtender*). Colormap designed by Thyng et al. (2016).

is the Stream Power with Alluvium Conservation and Entrainment (SPACE) component, which has been developed to simu-
late fluvial sediment transport and incision (Shobe et al., 2017) and is showcased here as an example of model coupling with
HyLands.

The integration capabilities of Landlab, where new and existing components can be combined in a straightforward way, opens
up new possibilities for applied environmental engineering and fundamental scientific research. For HyLands in particular, the
coupling of a deep-seated landslide algorithm with a sediment routing system will (i) help on a more applied level to explore
the impact of future changes in storm frequency on landslide occurrence and sediment dynamics (Fan et al., 2019), and (ii)
on a more fundamental level to facilitate the investigation of the interaction between landslides and sediment dynamics over
geological timescales. The latter is illustrated in Fig. 13, where we use the Landlab software to simulate the impact of uplifting
terrain on the formation of alluvial fans. Simulations are executed with and without landslide activity (Fig. 13a vs. Fig. 13b).
Resulting magnitude-frequency and area-volume relationships for the simulated landslides are shown in Fig. 14. The evolution
of the alluvial fans is further visualised in the movies listed in Table 4. For details regarding the algorithms and physics
supporting the HyLands component, see Campforts et al. (2020).



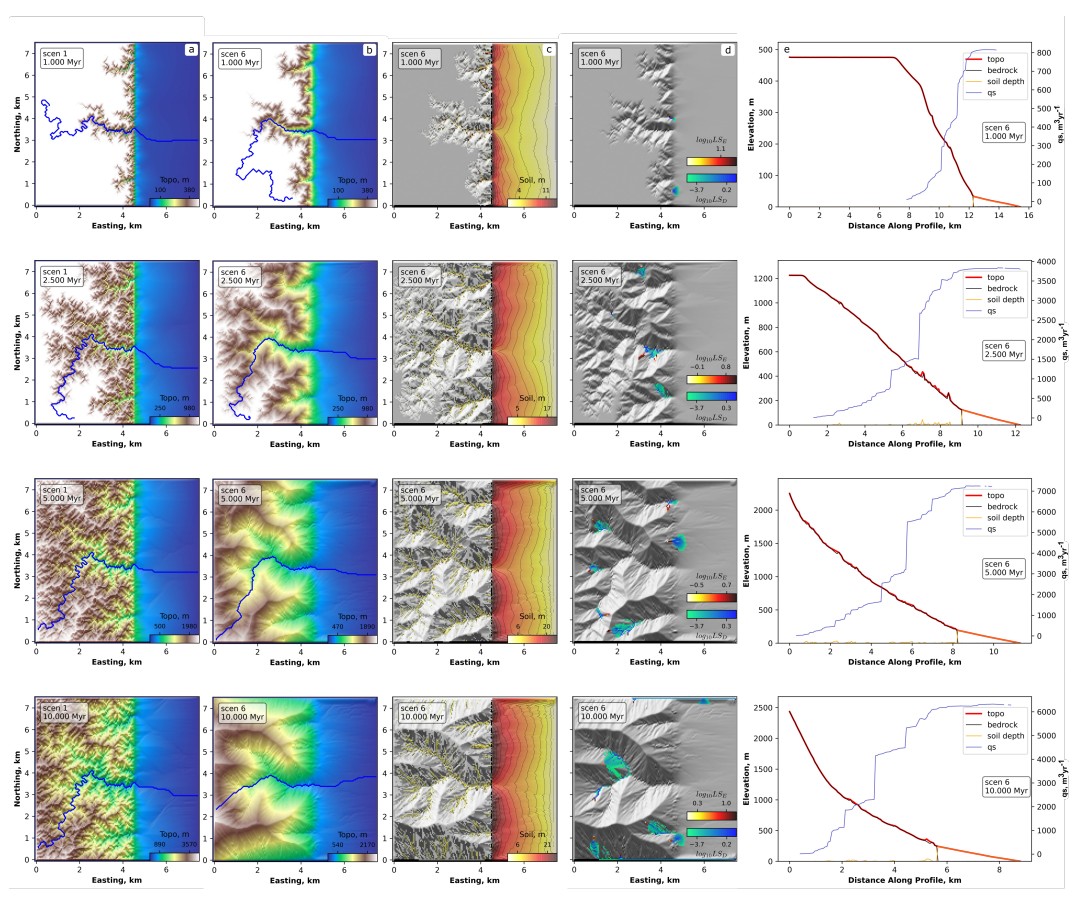

**Figure 13.** Illustration of HyLands model component. **a.** Time slices showing evolution of the landscape to steady state, without landsliding. The blue line represents the location of the river plotted in subplot (e). **b.** Same as (a), with landsliding. **c.** Sediment accumulation and the formation of alluvial fans. **d.** Landslide activity during the depicted time step. Red colors represent the logarithm of the landslide erosion, blue colors represent deposition. (e.) Shows the topographic and bedrock elevation (red and black line respectively) and the sediment thickness (orange line). The sediment flux is plotted against the right-hand y-axis (blue line). Note that, during landsliding, both pure landslide dams arise (red bumps on the profile) as well as irregularities in the bedrock profile (grey bumps). The latter originate from the river being redirected after landsliding, forming epigenetic gorges.

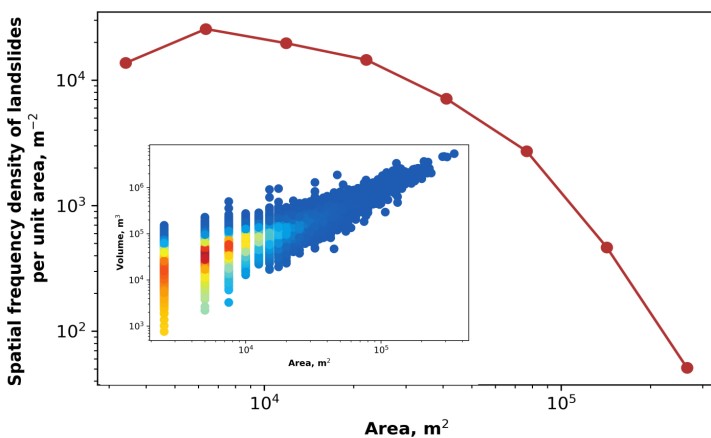

**Figure 14. (a)** Magnitude-frequency relationship of the landslides simulated with Landlab-HyLands (red dots) and illustrated in Fig. 13. **(Inset)** Scatter density plot showing the simulated Area-Volume relationship.

**Table 4.** Simulation movies created with Landlab-HyLands

| Scenario | Description | Link |
| --- | --- | --- |
| No landslides | topography | https://youtu.be/c5d7T8eehxw |
| | location of longest river | https://youtu.be/GqokukWi9cs |
| | river profile | https://youtu.be/A_JZ9POfJ54 |
| | sediment thickness | https://youtu.be/t0_tel5fhbM |
| Landslides | topography | https://youtu.be/1K_ceKYt9Nw |
| | location of longest river | https://youtu.be/YkmbUTN7zlI |
| | river profile | https://youtu.be/jyWgKTcMe74 |
| | sediment thickness | https://youtu.be/rwEBqGtHZs0 |
| | landslide erosion and deposition | https://youtu.be/_xoSm7p4ZxI |

## 6.6 Standard Names

Ensuring interoperability when coupling models or selecting datasets as inputs to models requires accurate alignment of scientific variables. Scientific variables are complex concepts composed of multiple facets—a phenomenon or object of observation, the corresponding physical quantity being measured, spatiotemporal context for the phenomenon, spatiotemporal reference for the measured quantity, mathematical operations applied to transform the physical quantity, etc. Because of this, and because terminology varies across disciplines, the semantic mediation task—determining whether two variables represent compatible concepts—can be quite involved. In CSDMS, BMI works in tandem with the CSDMS Standard Names (CSN) (Peckham



et al., 2013) to ensure proper alignment between resources. The Standard Names were developed to standardize and unify the
representation of scientific variables within CSDMS.

A CSDMS Standard Name contains two parts: an object part and a quantity part, with adjectives and modifiers (as prefixes) being used to help avoid ambiguity and identify a specific object and a specific, associated quantity. The quantity part may include one or more operation prefixes that create a new quantity from an existing quantity. An example related to surface-water hydrology is the runoff rate, for which the Standard Name is

`land_surface_water__runoff_volume_flux`

The double-underscore separates the object (surface water on land) from the quantity (the volume flux of runoff). The word "flux" implies a quantity per time per surface area, and so the implied dimensions are length per time.

As with all standard naming approaches, the Standard Names are limited in the amount of information they can represent because their data model and definitions are not explicitly represented. The Scientific Variables Ontology (SVO) (Stoica and
Peckham, 2018, 2019b, a; Stoica, 2020; SVO, 2020), a blueprint for representing scientific variables utilizing a compact set of domain-independent categories, relationships, and modular design patterns, was developed to address these issues. In computer science, an *ontology* is a system that attempts to capture and organize knowledge in a particular domain (in machine readable form), as understood by experts in that domain or subject area. In SVO, the CSN are represented with an explicit, formal model in machine-readable form using Semantic Web best practices (W3C Working Group, 2008). Because SVO is formalized, it
can be used to enable searching, semi-automated generation of new variable representations, and inexact but sufficient variable alignments through logical reasoning.

## 7   Community engagement

One of CSDMS' major activities has been the creation of a thriving community around Earth-surface dynamics modeling (Overeem et al., 2013). As of 2020, over 2,000 members, representing 552 institutions (144 US academic) and 71 countries, had
joined the community. CSDMS is, by design, a broad and deep coalition of members from disciplines reflected by five Working Groups and seven Focus Research Groups (Figure 15). From its inception, CSDMS has encouraged trans-disciplinarity by providing opportunities such as annual meetings, workshops, hackathons, and training events for domain scientists to interact with colleagues from other Earth and social science disciplines. These connections are essential for knowledge exchange among community efforts and allow for wider penetration of new technology and ideas. Cross-pollination of ideas from these events
and other community-member interactions have led to a variety of independently funded research projects. CSDMS has played a key role in shifting the paradigm to open code sharing in the Earth surface processes by facilitating resource sharing through model, data, and education repositories on the CSDMS web portal. CSDMS also offers a variety of services to community processes and the geoscience subdiscipline of interest. Along with their disciplinary expertise, researchers who work with computational models also need a strong foundation in programming, advanced computing, and data analytics (Atkins et al.,
2011).





Traditional Earth science education does not usually equip students with skills to use modern cyberinfrastructure and computing resources efficiently, or to become model developers (Campbell et al., 2013). The Earth surface processes community critically needs a platform to teach modern programming practices and high-performance computing methods to develop innovative models that can be used to understand and predict how the Earth's surface responds to environmental change and human influence. The practice of modeling lies at the core of predictive Earth-surface sciences, and educators should engage students in building, testing, and applying models (Hestenes, 1996; Manduca et al., 2008), but we found from a review of course catalogs that in practice the undergraduate curricula of more traditional discipline-focused departments do not include this component (Campbell et al., 2013). This issue is not entirely unique to Earth-surface sciences. The geosciences today are intensively quantitative, and there is an urgent need for a workforce with strong STEM skills (Council et al., 2012). The United States' National Science Foundation (NSF) recognizes as one of its "10 Big Ideas" that pathways are needed for educators to create a 21st-century workforce capable of effectively dealing with data (King and South, 2017). Moreover, an agile STEM workforce is considered a national priority (Atkins et al., 2011). Realizing this, CSDMS provides hands-on training opportunities during meetings. Some efforts are meant to build foundation—for example, via short courses that equip graduate students with skills in best programming practices. Other outreach efforts consist of short clinics, targeted to give potential users of cyberinfrastructure an active feel for certain models or computational techniques, or to provide an update to experts on new developments. More extensive separately organized hackathons bring together small science teams to work on solutions for more specific outstanding research problems. In 2020, CSDMS inaugurated an immersive Earth Surface Processes Summer Institute for students and early career scientists, focused on capacity building for Earth-surface processes modeling.

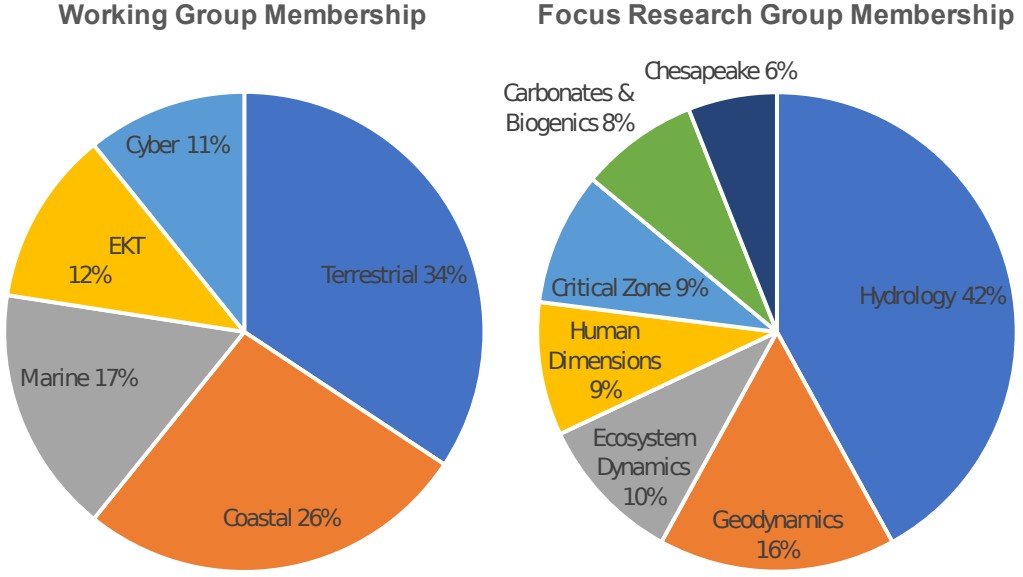

**Figure 15.** CSDMS Working Groups and Focus Research Groups.





## 8 Discussion and conclusions

In May 2020, the US National Science Foundation released a special report, prepared by the National Academy, on research opportunities in the Earth sciences (NRC, 2020). The report highlighted three unique types of research infrastructure: instrumentation, human infrastructure, and cyberinfrastructure. The report's recognition of cyberinfrastructure as a distinct form of research infrastructure is one indication of the critical role that computing now plays in the Earth and environmental sciences. Environmental modelling, and the software and culture-of-practice that support it, constitutes a key part of that cyberinfras-

tructure. Research software *is* infrastructure, and deserving of the same care and attention as a laboratory or field station. So too are the professional research software engineers who devote their expertise to helping the community do computational work more efficiently, effectively, and sustainably.

The research enterprise benefits when modeling software and tools are shared, coordinated, and interoperable, such that the six model operation tasks listed in Figure 2 can be done efficiently and effectively. For the Earth surface sciences, the CSDMS

Model Repository provides a community platform for finding and sharing model codes and related tools. In addition to acting as a valuable community resource, the Repository provides a solution to the growing mandate from journals and funding agencies to make research software openly available. The provision of standardized metadata and bibliographic information helps those who are looking for models to compare and evaluate the alternatives.

Simply providing source code and metadata is not enough, however. In order for Earth and environmental models to function

as community resources, they must be usable, and one of the key dimensions of usability is interoperability. The BMI standard promotes interoperability by reducing the learning curve for executing and querying models, and by greatly simplifying the process of linking (one way) or coupling (two way) models. A model program equipped with a BMI becomes an interoperable, standardized component: an element of an integrated system, rather than an idiosyncratic stand-alone product. One of the key abilities offered by a BMI-enabled model is run-time control, query, and modification. Because BMI supports step-wise

execution, a user can effectively pause a model in mid-run to inspect its state variables, and modify parameters or data. This capability allows iterative, loop-based coupling of models using simple scripts. The ability to query and modify values also enables tighter coupling. For example, if component models are treated as representing individual terms in a governing equation, a coupling script can use BMI functions to query each component's derivatives, construct a matrix, solve it, and then pass the updated state variables back to the individual components.

One advantage of BMI is that it is language agnostic, and can in principle be implemented in nearly any programming language. It can, for example, accommodate legacy codes written in Fortran. The disadvantage of language flexibility is that BMI addresses the least common denominator, and therefore does not take advantage of the more advanced features available in some languages, such as object-oriented capabilities. To some extent this disadvantage can be addressed by building more specialized, language-specific interfaces in parallel with a BMI. For example, Landlab components, which are implemented

as classes, use a lightweight, Python-specific interface that takes advantage of that language's object-oriented capabilities, advanced data types, and parameter-passing syntax. At the same time, Landlab also includes functionality to translate any of its components into a standard BMI component, so they can be integrated with components written in other languages.





The flexibility that BMI offers has led to its adoption in a variety of different applications, including US Geological Survey rainfall-runoff models (Markstrom et al., 2015; Regan et al., 2018, 2019), hydrodynamic modeling (including flagship models developed by Deltares and the Netherlands eScience Center, Hoch and Trigg, 2019; Hoch et al., 2019), delta and coastline evolution modeling (Ratliff et al., 2018), and modeling of methane emissions (Fox et al., 2020), to name a few. One disadvantage of a standard interface like BMI is the extra up-front investment in program development. Researchers may not perceive value in adding a standard interface to a legacy code, or writing it into a new code. However for codes whose scope merits repeated re-use, this effort usually more than pays for itself. Code written to a standard like BMI tends to be more modular and therefore easier to maintain. Existing templates for common languages in the Earth and environmental sciences make the process of providing a BMI to a new program is relatively painless: just a matter of filling in a set of pre-defined function names and signatures (Hutton et al., 2020). Adding a BMI to an existing legacy model can be a bit more involved, depending on how the program code is structured, because it often requires some degree of refactoring. Even in that case, we find that adding a BMI to a legacy model often makes that code more understandable and adaptable.

The variety of different programming languages used in the Earth and environmental sciences community presents a barrier to interoperability. The majority of models and tools in the CSDMS Repository are written in C, C++, Fortran, Python, and Matlab (Figure 4). Other languages used in CSDMS constituent communities include R (especially in ecosystem dynamics) and NetLogo's java-based scripting language (for agent-based modeling). Julia, a relatively new high-level language oriented toward numerical computing, also seems to be growing in popularity in the science community. Crossing the language barrier requires language-bridging tools. Translating the existing wealth of legacy code into a single, common language would be impractical, even if the community could agree on which language to use. A more effective solution is to *librarize* models and tools (Brown et al., 2014) as components that can be accessed and executing through a high-level scripting language. In CSDMS' case, the Babelizer tool provides this capability for codes written in C, C++, and Fortran, using Python as the bridging language.

Librarization can be applied to data sets too. The CSDMS Workbench accomplishes this with Data Components that provide function-call access to various data sets. Using the BMI syntax for data access removes the need to worry about data formats, and makes it easier to swap between data sets and models (for example, data versus model of ocean-wave properties) as components in a linked system. In this case, the BMI does not replace the more sophisticated data-access capabilities of a language-specific library like xarray, but it has the advantage of providing a consistent interface across multiple languages.

For building and modifying numerical models, the CSDMS Workbench provides Landlab as a Python-specific solution for 2D, grid-based applications. Experience with Landlab since its introduction has shown that a library of model "building blocks" can greatly reduce barriers on the software side of model creation. One indicator of the success of this approach is the growing number of Landlab-built models created by doctoral students as part of a larger body of dissertation research (e.g., Adams et al., 2017; Gray et al., 2017; Shobe et al., 2017; Lai and Anders, 2018; Langston and Tucker, 2018; Schmid et al., 2018; Strauch et al., 2018; Glade et al., 2019; Reitman et al., 2019; Carriere et al., 2020; Litwin et al., 2020). The ability to assemble models out of reusable "process components" allows for rapid construction of complete, multi-element models. One example of the value of rapid model assembly is a recent comparative testing and calibration study of long-term landform evolution models





(Barnhart et al., 2020b, c). The study authors used Landlab to develop a Python package for multi-model analysis of drainage basin evolution (Barnhart et al., 2019). The package allowed for the exploration and testing of more than 30 mathematically distinct models, as alternative hypotheses—a feat that would not have been possible with a traditional monolithic modeling code. This example illustrates how flexible, component-based modeling software promotes hypothesis testing.

Experience with BMI, pymt, and Landlab highlights the critical importance of documentation, consistent with the findings of Lawrence et al. (2015). Tutorial examples in particular provide a starting point that users can build on. Embedding tutorials in Jupyter Notebooks provides an effective way to combine descriptive text, program code, plots, and formatted mathematics. For reference-level documentation, document generator tools like Sphinx and doxygen translate internal documentation (comment blocks inside source code) into nicely formatted, web-accessible reference material.

A successful community cyberinfrastructure for numerical modeling requires more than just technology. It also takes community building and coordination. In the case of CSDMS, the community centers around common interest in a broader theme (Earth-surface processes) and a common approach (modeling). Activities such as meetings, workshops, hackathons, and webinars can help draw attention to new tools and methods, provide education in their use, and contribute to building a culture of resource sharing.

One of the biggest challenges to a fully functional community software ecosystem in Earth and environmental modeling is a lack of formal training in computational skills. Most geoscientists are self-taught programmers, and generally unaware of practices and tools that would make their work more efficient and sustainable. CSDMS and other community facilities have had some success in addressing this need with workshops, webinars, and summer schools, but there remains a need to scale up these efforts. Geoscience researchers should not need the equivalent of a computer science degree to perform computational research, but in our experience there is a basic set of skills that can make a big difference, yet which relatively few geoscientists possess. Potential solutions range from regular university-based, geoscience-oriented courses, to focused, community-led summer courses (like CSDMS' ESPIN, or the IsoCamp program for isotope geochemistry), to fully online courses. Questions of credit and funding inevitably arise, as does the issue of how to squeeze more material into already-packed curricula.

Another challenge revolves around incentives. The community as a whole clearly benefits from a FAIR and sustainable research software ecosystem. As noted above, the advent of software journals and peer-reviewed repositories (such as COMSESnet and pyopensci) provides one mechanism to encourage the creation of lasting digital products. The reproducibility movement provides another useful push, and has led journals and funding agencies to raise their standards for sharing and accessibility of software and other digital products. To take advantage of this momentum, hiring and promotion committees at universities and research organizations need to acknowledge the value of contributions to high-quality research software. Professional societies can contribute by offering awards that recognize contributions to cyberinfrastructure.

The third major challenge is support. Our experience with CSDMS demonstrates that a modest investment in community-oriented computing can have a substantial positive impact on research productivity. By investing in stable community repositories, interoperability standards, and software libraries and frameworks, a funding agency can increase the impact of its portfolio by incentivizing a shared, reusable and ever-improving community infrastructure of models, tools, and expertise. A key to mak-




ing this approach scalable, in addition to incentives, is to provide sufficient documentation and consulting support to enable community members to create research cybertools that are Findable, Accessible, Interoperable, and Reusable. We have found from our own experience that consulting support is an especially important piece. Projects that include a professional research software engineer in their team—even if it is just at the level of general design advice, informal education, or help overcoming technical obstacles—tend to be much more likely to produce robust, flexible, sustainable software as a lasting broader impact of a project.

Computational modeling in the Earth and environmental sciences has come a long way since the dawn of the 3rd millennium.
The possibilities of a coordinated, community-wide cyber-ecosystem are starting to emerge. Fully achieving this vision will require a combination of education, incentives, and support. Universities, research agencies, and individual researchers all have a role to play.

*Code availability.*

The current versions of the various elements in the CSDMS Workbench software suite are available under the MIT licence.
As of this writing, Landlab code, documentation, and tutorials are available in a git version-control repository on the GitHub hosting site at https://github.com/landlab/landlab. Documentation can be accessed at https://landlab.github.io. The Landlab version discussed here is 2.0 ("Mrs. Weasley"), available via Zenodo at https://doi.org/10.5281/zenodo.3776837. Current versions of software, technical specifications, documentation, and other resources for other Workbench elements (BMI, babelizer, pymt, model and data components) are managed on GitHub under the CSDMS organization (https://github.com/csdms). On-
line documentation for BMI, pymt, and babelizer is presently hosted on the Read the Docs platform (for example, https://bmi.readthedocs.io, and similarly for pymt and babelizer). The BMI version presented in this paper is 2.0, available via JOSS and Zenodo at https://doi.org/10.21105/joss.02317. Babelizer version 0.3.8 can be found at https://doi.org/10.5281/zenodo.4985181. Version 1.3.1 of the Python Modeling Toolkit (pymt) can be accessed at https://doi.org/10.5281/zenodo.4985222. The simulation shown in Figure 12 is contained and described in two Jupyter Notebooks available at https://github.com/gregtucker/
component_modeling_csdms_chapter.

*Author contributions.* CSDMS is the outcome of a community-wide effort, with contributions from numerous community members to governance, workshops, software, educational resources, and ideas. In terms of this particular manuscript, primary author contributions by section were: Sections 1–4 (GT), 5 (AK), Section 6 (MP, EH, TG, BC, KB, GT, SP), Section 7 (IO, LM), Section 8 (GT). JS founded CSDMS and led it for 10 years. All authors contributed to editing the manuscript.

*Competing interests.* The authors declare that they have no conflict of interest.



*Acknowledgements.* The Community Surface Dynamics Modeling System (CSDMS) is supported by the US National Science Foundation (NSF) (1831623). Initial development of Landlab was supported by the NSF SI2 program (1450409). The Permafrost Toolbox was supported by NSF OPP (1503559). ESPIn is supported by NSF-CISE (1924259). Additional sources of support include EarthCube (2026951), and an NSF postdoctoral fellowship (1725774 to KRB). The authors gratefully acknowledge the contributions of numerous CSDMS members, whose service on committees, sharing of codes, teaching of clinics, and other efforts have created a vibrant community of practice.



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
