# Peer review of "CSDMS: A community platform for numerical modeling of Earth-surface processes"

_Geoscientific Model Development, 2021_

## Author Comment (AC1)

We thank the reviewers for their thoughtful feedback, and the editorial staff for managing this manuscript. Responses to specific reviewer comments are given below. The original comments are in plain text, and our responses are in **bold**.

**Response to Reviewer 1 Comments (RC1)**

**We thank Reviewer 1 for taking a careful, thorough look at the manuscript, with helpful comments that range from interesting high-level issues (e.g., verification and validation) to typographical consistency.**

**General Comments**

This paper has two purposes – to inform the reader about activities of the CSDMS and to describe the tools and standards developed for model interoperability.

1. On the usage of FAIR. FAIR standards apply to data not software. Several papers have discussed why software is not data which motived the development of FAIR4RS. While FAIR is used appropriately in some contexts, it is not used appropriately in others.

   **To address this issue in the manuscript, we have added text to the third paragraph of Section 5 to note that FAIR principles are a bit different when applied to software. We added citations to three recent papers that delve into this topic (we also cite them earlier as well, when referring briefly to FAIR, so that readers understand we are talking about their adaptation to research software).**

2. Verification and Validation. The paper begins a discussion on code testing but does not address this point further for codes in the CS-DMS repository. Assuming unit testing is done, and codes are validated and verified, it would be nice to have a discussion on what this means in the context of the ecosystem created especially in coupling models/components and error propagation.

> **This is a great idea, and we have added a paragraph to the Discussion section addressing this point. The specific issue of error propagation is a bit beyond the scope of this paper, because it is so specific to particular models and systems. But the general issue of verification testing is indeed important, and this new paragraph discusses the current state of practice and our experiences vis-a-vis testing.**

3. Computational Overhead. CSDMS has created a rich ecosystem of tools and standards. Has any performance benchmarking been done to understand the computational overhead?

> **This is a hard question to answer in general because there are so many different types of models in the Repository. Some are trivial to run, some cannot be used effectively without an HPC system, and many are in between in terms of performance. Even within more focused and centrally coordinated tools such as Landlab, there exists a spectrum in terms of performance. However, we like the reviewer's suggestion of discussing performance, and to that end have added a paragraph to Section 6.5 discussing performance issues in the context of Landlab specifically.**

**Specific Comments**

Line 94: I am unclear what the analogy the quote is trying to draw.

**Sentence re-worded.**

Line 143: The reviews for JOSS journals are objective reviews set against a checklist of items. Reviewers check for what the authors says it does. Do they inspect the software code itself, that is "evaluate the software directly"?

**We re-worded the sentence to be more specific about what a JOSS review entails.**

Figure 2: Include a descriptive figure caption.

**We added descriptive text.**

Line 241: Strictly speaking, reproducing needs an executable, not necessarily

the source code.

**We agree, though for transparency source code is better. We have modified the sentence accordingly.**

Line 244: FAIR data need a persistent identifier such as a DOI.

**Yes, a DOI is just one example of a persistent identifier; we have re-worded this.**

Figure 3. Include a descriptive figure caption.

**We added a descriptive caption.**

Line 347: What happens in the interpolation when there is a first order discontinuity?

**Grid interpolation is associated with a number of technical issues, including handling of discontinuities, strengths and weaknesses of alternative interpolation methods, etc. These are at a level of detail that lies beyond the scope of this manuscript. However, we have added a citation for the grid remapping package so interested readers know where to look for answers to questions such as this one.**

Line 360: What happens when the timesteps are not multiples of one another?

**We have modified the example here to make it clear that timesteps do not need to be multiples of one another.**

Figure 13: Clarify that c. and d. are with the "with landsliding" model

**We added this to the caption for (c) (case (d) already includes the word landsliding).**

Line 580: Sentences is awkward. Librarization of tools to access data sets not librarization of the data sets themselves.

**Good point, we re-worded this sentence accordingly.**

**Technical Corrections**

Review usage of title case in section and subsection headings for consistency

**Headings are now lower case except the first letter and any proper nouns.**

For inline text, use Fig. or Figure consistently. Similarly for figure referenced in parens, ()

**We changed Figure to Fig. throughout except where the word "figure" starts a sentence.**

Figure 1: Many of the labels on the figures are hard to read at 100% magnification

**We have edited Figure 1 to enlarge some of the labels, and remove others that were unnecessary to the point of the figure. Some fine-print text is embedded in the originals and hard to remove, but our hope is that the modified version gets the intended point across that Earth surface dynamics models range widely in their domains and time scales.**

Line 154:

For citing PETSc, see:

ftp://ftp.mcs.anl.gov/pub/petsc/nightlylogs/xsdk/xsdk-configuration-tester/packages/petsc/src/do

**References added.**

deal.II (not capitalized)

**Fixed.**

For citing deal.II, see:

https://www.dealii.org/publications.html

**References added.**

Line 279, 359: Are contractions allowed?

**We couldn't find this in the guidelines, so will leave it for the copyeditor to recommend one way or the other.**

Line 295: References Fig. 6 before referencing Fig. 5 (and not referenced elsewhere)

**Figure ordering fixed.**

Table 3. Change Table caption to: ". . . Python Modeling Toolkit pymt."

**Correction made.**

All models are capitalized here but may be used as lowercase in text. Theses should be consistent.

**Table modified for capitalization consistency.**

Line 384: (Fig. 2). Should this be deleted?

**Agreed that this figure reference does not add much; deleted.**

Figure 1 and Figure 13: Descriptions use a syntax of referring to the different panels e.g,. a. vs (a)

**Figure 13 caption modified for consistency.**

Figure 13: Sediment thickness is plotted as soil depth as a yellow and not an orange line according to the legend.

**It is actually an orange line, just a rather "yellow-ish" orange.**

Figure 14: Technically there is no "(b)" so should there be an "(a)"?

**Good catch, letter removed.**

Figure 15: Define EKT

**Acronym definition added to caption.**

Line 614: Define ESPIN

**The future of ESPIN is presently uncertain, so we removed the examples here.**

Line 618: CoMSES Net

**Corrected.**

Line 619: pyOpenSci

**Corrected.**

**Reviewer 2 Comments (RC2)**

**We thank Reviewer 2 for their thoughtful reading and generous feedback.**

This manuscript outlines the challenges, opportunities, and contributions of the Community Surface Dynamics Modeling System (CSDMS) community towards advancing integrated, extensible, and reproducible modeling of Earth surface processes. The authors provide some conceptual context for the problem and issues that they are addressing, which is the increasing importance of models in Earth surface processes, in conjuction with some of the challenges that this growing and evolving facet of Earth science is experiencing. The authors outline an interesting set of model operations – drawing a comparison with Bloom's Taxonomy – of increasing complexity that are required of models. They then review a suite of tools developed in support of the CSDMS effort that address varying aspects of this taxonomy.

The manuscript is very well written and organized. I am particularly fond of their proposed taxonomy and believe that could be the basis of further expanding how we not only think of models, but also how we thinking about educating scientists about models. I presents relevant examples using tools within the CSDMS ecosystem, as well as an overview of the ways that CSDMS seeks to promote development and acquisition of modeling skills and habits of mind in the community.

In all honesty, I cannot find any significant errors or issues within this manuscript. My only very minor quibble is that I believe that the authors could perhaps elaborate more on some of the more future-oriented challenges, particularly as they relate to educating the next generation of scientists. The authors, in the Discussion, allude to a set of skills that CSDMS has observed are important for modeling and I wonder if those could

be posed as learning outcomes that might be necessary for students. Could, for example, figure 2 be replicated along with some key skills or learning outcomes that are appropriate to each of these operations. To be clear, I don't think the authors need to address this suggestion, as it may be beyond the scope of this manuscript, but that is what immediately came to mind when I read the article.

**We're happy to hear that the taxonomy idea resonated, and like the idea of using it as a framework for identifying key skills. Although doing so in a detailed and in-depth way is beyond the scope of this manuscript, we have added a few sentences to the Discussion to point out that the taxonomy could provide guidance for designing learning goals, and identified a few potential topics as examples.**

I believe this manuscript makes an important contribution to the Earth surface modeling community and suggest that it be published in its present form.

**Thank you!**

---

## Author Response (AR2)

**Responses to Topical Editor comments**

January 13, 2022

(Original comments in plain text below; our responses in **bold**)

Based on the positive reviews and your responses to them, I am happy to recommend your paper for publication, pending a couple small questions of my own on your changes to the manuscript in response to the referees' comments. Line numbers are from the tracked-changes version.

**We thank the topical editor for catching these! We have made edits to the relevant lines as noted.**

Line 245: I am not sure that you meant to use the word "code" before the parentheses; this seems in response to the referee comment about reproducibility (vs. transparency, as you note) in which the referee noted that an executable would suffice for reproducibility.

**Changed to "... requires shared digital files (either executable binary files or source code; ideally the latter so that the algorithms are transparent)"**

Line 635: "failures Jupyter" – it seems that a sentence transition was lost during the editing process.

**Changed to "... run-time errors in ..."**

Line 676: "dawn of the 3rd millennium." Just :)

**Admittedly we might still be in that dawn, so changed to "... in the first two decades of the 21st century"**

Line 698: Thank you for thanking the reviewers!

**Added a thank you note to the editorial staff too!**